# Long-Term Outcomes of in Utero Ramadan Exposure: A Systematic Literature Review

**DOI:** 10.3390/nu13124511

**Published:** 2021-12-17

**Authors:** Melani R. Mahanani, Eman Abderbwih, Amanda S. Wendt, Andreas Deckert, Khatia Antia, Olaf Horstick, Peter Dambach, Stefan Kohler, Volker Winkler

**Affiliations:** 1Heidelberg Institute of Global Health, Heidelberg University Hospital, Im Neuenheimer Feld 130.3, 69120 Heidelberg, Germany; melani.mahanani@uni-heidelberg.de (M.R.M.); eman.abderbwih@uni-heidelberg.de (E.A.); a.deckert@uni-heidelberg.de (A.D.); khatia.antia@uni-heidelberg.de (K.A.); Olaf.Horstick@uni-heidelberg.de (O.H.); peter.dambach@uni-heidelberg.de (P.D.); stefan.kohler@uni-heidelberg.de (S.K.); 2Potsdam Institute for Climate Impact Research, Member of the Leibniz Association, P.O. Box 601203, 14412 Potsdam, Germany; wendt@pik-potsdam.de

**Keywords:** Ramadan, in utero, pregnancy, systematic review

## Abstract

Health outcomes of in utero Ramadan exposure have been reported in a systematic literature review; however, the available literature on long-term effects were not fully covered. Our study aims to specifically review the long-term outcomes of in utero Ramadan exposure. We searched for original research articles analyzing any long-term outcome of in utero Ramadan exposure, excluding maternal and perinatal outcomes. Sixteen studies from 8304 non-duplicate search results were included. Most studies suggest negative consequences from in utero Ramadan exposure on health, as well as on economic outcomes later in adulthood. Higher under-five mortality rate, higher mortality under three months, and under one year, shorter stature, lower body mass index, increased incidence of vision, hearing and learning disabilities, lower mathematics, writing and reading scores, as well as a lower probability to own a home were associated with Ramadan exposure during conception or the first trimester of pregnancy. Furthermore, age and sex seem to play a pivotal role on the association. Existing studies suggest that in utero Ramadan exposure may adversely impact long-term health and economic well-being. However, evidence is limited. Meanwhile, increasing awareness of the potential risks of Ramadan fasting during pregnancy should be raised among pregnant women and clinicians and other antenatal care workers should promote better maternal healthcare.

## 1. Introduction

During Ramadan, healthy Muslim adults are obliged to fast from dawn until sunset, abstaining from food, drink, medicine, and sexual activity. Not only is the timing of meals altered during the Ramadan period, but the type of food eaten, as well as sleeping and behavioral patterns, are substantially changed for about 4 weeks [1]. Improvements in dietary diversity during Ramadan were found even when Muslims did not adhere to fasting [2]. A growing body of evidence has identified direct and indirect health effects of Ramadan fasting in adults, such as changes in lipid, carbohydrate, protein, and hormone metabolisms, as well as in body weight [1,3,4]. Individuals’ moods and irritability also appear to change during Ramadan, which has been associated with restricted nicotine use and reductions in sleep [5]. Overall, health effects among Ramadan fasting adults have been described as manageable or even beneficial [1,6].

In contrast, the effects of Ramadan on the unborn child are less explored. Even though Islamic law gives permission for pregnant women to opt out of fasting, it is common that expectant mothers follow Ramadan fasting due to different reasons, such as to improve their spiritual activity [7], as well as due to influence from partners and families [8,9]. It is widely perceived that the early phase of pregnancy, when organogenesis takes place, is crucial for human development [10]. During the first trimester, when the immune system, endocrine and metabolic pathways develop, the fetus may be particularly prone to effects from Ramadan fasting, such as short-term undernutrition [10,11]. An analysis of in utero Ramadan exposure as a natural experiment showed a significantly elevated mortality among children under five [12]. In principle, a natural experiment makes use of an individual’s assignment to the experimental and control conditions by nature or by other factors outside the control of the investigators. However, a systematic literature review on the perinatal outcomes of Ramadan fasting during pregnancy was inconclusive [13].

In contrast to short-term outcomes, our study aimed to collect and evaluate existing evidence on the relationship between in utero Ramadan exposure and its long-term effects, excluding perinatal and maternal outcomes. A recent systematic review by Oosterwijk et al. [14] analyzed the impact of Ramadan exposure on health, including long-term effects, and concluded that no study reported significant long-term effects. However, the systematic review included only part of the available literature, and focused on health only.

Long-term effects of in utero famine have been well studied and showed an increased risk for obesity, diabetes, coronary heart disease, breast cancer and cognitive decline later in life [15,16,17]. Ramadan fasting is less dramatic than famine; however, it may have effects on the developing fetus later in life.

## 2. Materials and Methods

This systematic review was conducted in accordance with the Preferred Reporting Items for Systematic Reviews and Meta-Analyses (PRISMA) 2009 guidelines (see Appendix A) [18]. Two authors (M.R.M. and E.A.) independently performed the literature search, screened the titles and abstracts, assessed the full texts, and performed the data extraction and quality assessment. Any arising disagreement was solved by discussion among all authors until consensus was reached.

We searched the scientific literature utilizing the following databases from inception to 21 November 2021: PubMed, PsycINFO, EconLit, Index Islamicus, Web of Science, Cochrane Library, WHO Global Index Medicus, WHO Virtual Health Library, and Google Scholar. The searches were performed very broadly, using only the term “Ramadan”, which covers all common variations of Ramadan fasting because they all include this term. We included only original research articles that mainly assessed the long-term outcomes of in utero Ramadan exposure, and excluded studies reporting on either maternal or fetal outcomes. For Google Scholar, we modified the search strategy by screening 50 results stepwise after the first 200 records, until no further relevant studies were found. After identifying eligible articles, we searched their reference lists for additional papers. Full search strategy for all databases can be found in Appendix A.

We used the standardized data collection form of the Cochrane Collaboration Public Health Group [19] to extract the following information from each study: authors, publication date and journal, country, type of study, aims and objectives, sampling techniques and dates of data collection, sample size and age and sex of participants, exposures and outcomes including outcome measures, key conclusions, limitations and recommendations.

Afterwards, we used the Specialist Unit for Review Evidence (SURE) tools [20] to assess the quality of included studies (see Appendix A). Specifically, we applied the tool for cohort studies, which consists of the following 13 items: study design, research question, setting and location, participant selection, participant characteristics, exposures and outcomes measurement, bias consideration, study size, statistical methods, participant flow, results, sponsorship/conflict of interest and limitations.

We analyzed studies according to the following information: outcome measures, age, and sex of the population of interest. Considering the limited number of studies and a variety of outcome measures, we decided to forgo a meta-analysis.

## 3. Results

We identified a total of 8690 original research articles, of which 8304 remained after the removal of duplicates from the search results. Based on title and abstract screening, 8280 studies were excluded because they mostly focused on maternal or perinatal outcomes or on fasting adults. We screened the full text of 24 studies, of which 16 met the inclusion criteria (see Figure 1). Three studies were excluded because they did not present original research (commentaries), three focused on short-term consequences, one focused on the impact of religion adherence, and two studies were published twice in different journals with different titles; therefore, we only kept the primary publication. 

### 3.1. Study Characteristics

Six studies were conducted in South East Asia (all in Indonesia) [21,22,23,24,25,26], one in South Asia (Pakistan) [27], two in Eastern Mediterranean region (Iran) [28,29], two in Africa (one in Burkina Faso [12] and one in Ethiopia [30]) and three in Europe (one in England [31] and two in Denmark [32,33]). Two studies had a cross-regional focus including 39 countries in Eastern Europe, Africa, Eastern Mediterranean, Asia, and North America [34,35].

All studies used retrospective longitudinal data for analysis. Six manuscripts used nationally representative data from the Indonesian Family Life Survey (IFLS) [21,22,23,24,25]; one used data from the district-based Punjab Multiple Indicator Survey (MICS) [27]; two used school register data [28,31]; one used data from the Urban Health Equity and Assessment Response Tool (Urban HEART) [29]; two used Danish administrative records [32,33]; one study used Michigan (United States) natality data and census data for Iraq and Uganda [34]; one used data from the Nouna Health and Demographic Surveillance System in north-western Burkina Faso [12]; one used data from the Ethiopia Demographic and Health Survey [30]; and one analyzed data collected from 98 demographic and health surveys [35].

Studies investigated the children of Muslim parents [21,23,30,32,33,34], children of Ramadan-exposed Muslim mothers (experiencing Ramadan during pregnancy, with unknown fasting status) [21,22,24,25,26,27,29,31,35], and children of Ramadan fasting mothers [28]. All studies included non-exposed control groups represented by children of non-Muslim parents [21,23,30,32,33,34], children of non-exposed mothers (no Ramadan during pregnancy) [21,22,24,25,26,27,29,31,35], and children of non-fasting Muslim mothers [28]. Most studies targeted more than one age group, although some others targeted only children [27,29,31], adolescents [32], and adults [22,34].

Most studies [12,21,22,23,24,25,26,27,29,30,31,33,34,35] used an intention-to-treat design by estimating the Ramadan exposure based on an individual’s date of birth, calculating if Ramadan overlapped with the pregnancy, assuming a full-term birth (individual’s mother’s fasting status unknown). One study used questionnaire data asking mothers about fasting throughout their pregnancies, defining in utero Ramadan exposure as fasting for at least 27 days during pregnancy [28]. Another study defined in utero Ramadan exposure as when the mother and the child had immigrated from a country with a more than 90% Muslim population (individual’s fasting status unknown) [32]. Some studies highlighted that their study design allowed them to analyze Ramadan exposure during pregnancy as a natural experiment, indicating the possibility for causal inference [12,21,24,25,28,31,32,33].

Standardized tools for anthropometric measurements assessed by trained nurses in other studies included height-for-age z-scores [26,27,29,35], weight-for-age z-scores [26,30], and body-mass-index-for-age z-scores [26]. The Wechsler Intelligence Scale for Children-Revised (WISC-R) and the Wechsler Preschool and Primary Scales of Intelligence (WPPSI), conducted by two randomly assigned persons that did not know the exposure condition, were used to estimate individual’s intelligence quotient [28]. One study used self-reported information on breathing difficulties, and respondents were asked to indicate whether they had experienced wheezing or shortness of breath during the 4 weeks before the interview [21]. Another study took measurements of a diverse set of health variables such as weight, height, blood pressure, pulse, lung capacity, and hemoglobin level by nurses and applied the Nine-point General Health Scale for measuring general health by other health professionals. This study also used self-reported information on chest pain and slow-healing wounds [23]. One study used teacher assessment scores on the subjects of mathematics, reading, and writing [31]. A Danish study used standardized test scores from the national exams on the subjects of Danish, English, mathematics, and science administered by the Ministry of Education [32]. Two studies used the Raven’s Colored Progressive Matrices (CPM) for evaluating cognitive ability, i.e., general intelligence, without giving detailed information about the outcome assessment [24,25]. The same studies used data on mathematics scores, earnings, hours worked, employment, and child labor from household interviews, whereas the employment indicator, salary income, and annual hours were based on official reports by employers [24,25]. One study examined the educational attainment: whether a child whose age was between 7 and 11 was currently enrolled in a primary school, and whether a child whose age was between 15 and 20 had graduated from a primary school [30]. The wage was calculated on the basis of salary income divided by hours of work in that year [33]. One study used data on disability, home ownership, and employment status from National Censuses without further information regarding the outcome assessment [34].

Most authors conducted descriptive analysis to compare health and other outcomes between the exposed and non-exposed groups by calculating percentages, means with standard deviation (SD), 95% confidence intervals (95% CIs) and *p*-values. For further analyses, bivariable and multivariable models were performed, utilizing quantile regression [24], logistic regression [21,26,27,31,32], linear regression (OLS) [22,23,25,28,29,30] and kernel-weighted local polynomial regressions [35]. In most studies with a natural experiment design, difference-in-differences analyses (DID) [21,31,32,33,34] were performed.

Table 1 summarizes the main characteristics of all included studies, along with the quality assessment.

### 3.2. Study Results According to Outcome Measures

Table 2 provides an overview of results presenting all outcomes with statistical measures.

### 3.3. Health Outcomes

Adult Muslims who had been in utero during Ramadan were thinner and had a smaller stature than Muslims who had not, whereas among non-Muslims, no differences were found [22]. In line with these findings, another study showed that children born to religious Muslim mothers and exposed to Ramadan in the first trimester of pregnancy were shorter in late adolescence (15–19 years of age) compared with their unexposed siblings. Lower BMI was also observed and peaked in early adolescence (10–14 years of age) for exposed Muslim children in comparison to their unexposed siblings [26]. One study concluded, on the one hand, that under-five children exposed to in utero Ramadan were more likely to be stunted if the exposure happened in any month of gestation prior to the eighth month of pregnancy. On the other hand, children under five who were exposed to in utero Ramadan in the ninth month were taller and heavier than non-exposed children [27]. Another study supported this finding, showing that in utero Ramadan exposure was associated with decreases in children’s height at ages of 10 years or older [29]. In a cross-regional study of 37 countries, Ramadan exposure during early and mid-gestation affected the height of 3- and 4-year-old male Muslim children, and the effect tended to be stronger in West Africa, Central Asia, and other countries with a higher proportion of Muslims in the population. Effects on height in female children were not detected consistently [35].

When compared with children of non-Muslim mothers, the mortality under three months and mortality under one year of children born to Muslim mothers were higher when Ramadan occurred during the first trimester [30]. Similarly, another study found that the under-five mortality rates of children born to Muslim mothers were higher when Ramadan occurred during conception, and the first and second trimesters [12]. Vision, hearing, and learning disability incidence increased among Muslim adults when Ramadan occurred during the first month of pregnancy [34]. Using the WPPSI for children aged 4 to 6 and the WISC-R for those aged 6 and 13, fasting during pregnancy had no effect on the intellectual development [28]. In contrast, a study that analyzed Pakistani and Bangladeshi immigrants in England using student register data found that test scores on the mathematics, reading, and writing abilities of 7-year-old children were lower when exposed to Ramadan in early pregnancy [31]. Similarly, in Indonesia, another study identified lower scores on Raven’s CPM tests and lower mathematics scores among 8- to 15-year-old children who were in utero during Ramadan [24]. Using standard test scores drawn from Danish administrative records, a study showed that fetal exposure to Ramadan had a negative impact on the academic proficiency of Muslim students (aged 16 years), especially females [32]. In line with these findings, one study assessed that children exposed to Ramadan in utero scored lower on Raven’s CPM tests and on mathematics tests [25] than unexposed children.

In comparison with non-exposed Muslims, the risk of experiencing a symptom of any breathing difficulty (wheezing, shortness of breath) or being diagnosed with a lung disease (asthma or other lung condition) was higher among exposed Muslims. Wheezing prevalence was also higher among exposed adult Muslims living in Muslim areas and the association tended to increase with age, being strongest among those aged 45 years or older [21]. Exposure to Ramadan fasting before birth was associated with poorer general health, increased risk of slow-healing wounds, and chest pain (symptom indicative of diabetes and coronary heart disease). On average, exposed Muslims had a higher pulse pressure than non-exposed Muslims [23].

### 3.4. Economic Outcomes

Investigating health and economic consequences of in utero Ramadan exposure, one study showed that men exposed during first month of gestation were less likely to own a home when compared with non-Muslims [34]. In a Caribbean study, adults aged between 24 and 55 had a lower likelihood of employment if they were exposed to Ramadan around the seventh month of gestation in comparison with non-Muslims [33]. In Indonesia, a significant reduction in hours worked was found among exposed females aged 18–65 years. No significant findings were observed among exposed adult males [24]. Similarly, another study showed that children exposed to Ramadan in utero performed more child labor, and exposed adults worked fewer hours per week and were more likely to be self-employed than unexposed adults [25].

Several health and economic outcomes presented in Table 2 and described above indicate that being exposed to the month of Ramadan during conception or the first trimester of gestation was associated with higher under-five mortality rate [12], higher mortality under three months and under one year [30], shorter stature [26], lower body mass index [26], increased incidence of vision, hearing and learning disabilities [34], lower mathematics, writing and reading scores [31], as well as being less likely to own a home [34].

## 4. Discussion

The 16 available studies for this systematic review suggest that in utero Ramadan exposure may have negative long-term consequences on health and economic outcomes. Some negative effects were particularly observed when Ramadan occurred during conception or the first trimester of the pregnancy. Findings also suggest that the age at which the outcome is measured [21,26] and sex [24,35] may play important roles. These results are in line with the fetal programming theory, where during the first trimester, the fetus is in its most vulnerable state and particularly prone to negative effects from the surrounding environment. These effects are expected to manifest at post-reproductive age [21]. The strongest effects may also be due to higher fasting rates in early pregnancy, which survey-based studies have reported [36,37].

The British Islamic Medical Association has published a compendium of evidence on Ramadan fasting, and recommends avoiding Ramadan fasting in the first trimester, or intermittent fasting if the pregnant women are very keen to fast. Fasting in the second and third trimester of pregnancy seems to be relatively safe in healthy women (low/moderate-risk category). However, women with underlying health conditions should be fully assessed by a qualified healthcare professional to evaluate and categorize their level of risk (moderate/high/very high). Those in the very-high-risk group must not fast, and those in the high-risk group should be discouraged from fasting [38].

This systematic literature review is, to the best of our knowledge, the first to assess the available scientific knowledge about the long-term effects of in utero Ramadan exposure on health and economic outcomes. Even though almost all studies show small negative effects, it is difficult to generalize these findings due to a number of reasons: although we used a broad search strategy, we found few published studies, and these studies looked at many different outcomes. Therefore, we were neither able to perform a meta-analysis to strengthen the evidence, nor were we able to systematically tackle the possibility of publication bias. However, 13 out of 16 (81%) of our studies had samples over 10,000; thus, the case of included small studies being more likely to be “negative” may not be an issue in this work.

Another limitation is the retrospective nature of the studies. Prospective studies assessing Ramadan exposure more objectively are needed. Given the design of most studies using date of birth as the only indicator to define the in utero Ramadan exposure (intention-to-treat design), it remains unclear if the observed effects are due to Ramadan fasting itself or due to behavioral changes or any other event strictly related to Ramadan. Usually defining the exposure by date of birth tries to take into account some variability in the length of pregnancy by additional categorization with different cut-off values; however, it remains a limitation [23]. Furthermore, only one study directly assessed the fasting adherence of mothers. Therefore, pregnant women should also pay attention to the quality of their diet, stress levels, and sleep schedule during Ramadan [39]. The intention-to-treat design is prone to nondifferential misclassification and underestimates the real strength of the associations [12,21]. It has the benefits of the easy utilization of population-wide datasets for analysis, avoiding confounding due to non-random dropouts. Studies directly comparing Muslims and non-Muslims are prone to confounding, because over time, there may be considerable factors associated with the disease and with being a Muslim. The included studies mostly adjusted for age, sex, and month of birth.

## 5. Conclusions

In conclusion, this systematic literature review found limited evidence that, in line with the biological hypothesis, in utero Ramadan exposure may adversely impact long-term health and economic well-being. In some regions, up to 90% of pregnant Muslims practice Ramadan fasting [9], and in each generation, the global estimated number of up to 535 million babies are exposed to Ramadan in utero [13]; therefore, we highly encourage public health researchers to further explore possible impacts of in utero Ramadan exposure.

The aspirations of pregnant Muslim women who want to fast during Ramadan should be respected. At the same time, awareness of the potential risks of Ramadan fasting during pregnancy needs to be raised among pregnant women, equally with the risk of gestational diabetes mellitus and other fetal–maternal risks [40]. Clinicians and other antenatal care workers should promote better maternal healthcare to help manage a healthy pregnancy and reduce negative effects for their offspring [26].

On an individual level, disadvantageous consequences are likely to be relatively small compared to risk behavior, or may possibly even be due to bias. However, even small effect sizes are important from a public health perspective. Further research utilizing longitudinal data from the wider Muslim population is needed to generate clear scientific evidence and to determine the possible effects of in utero Ramadan exposure with respect to the three trimesters, the duration of the exposure, and the underlying cause. Thus, well-designed studies investigating Ramadan exposure during pregnancy are needed to investigate the full impacts on offspring.

## Figures and Tables

**Figure 1 nutrients-13-04511-f001:**
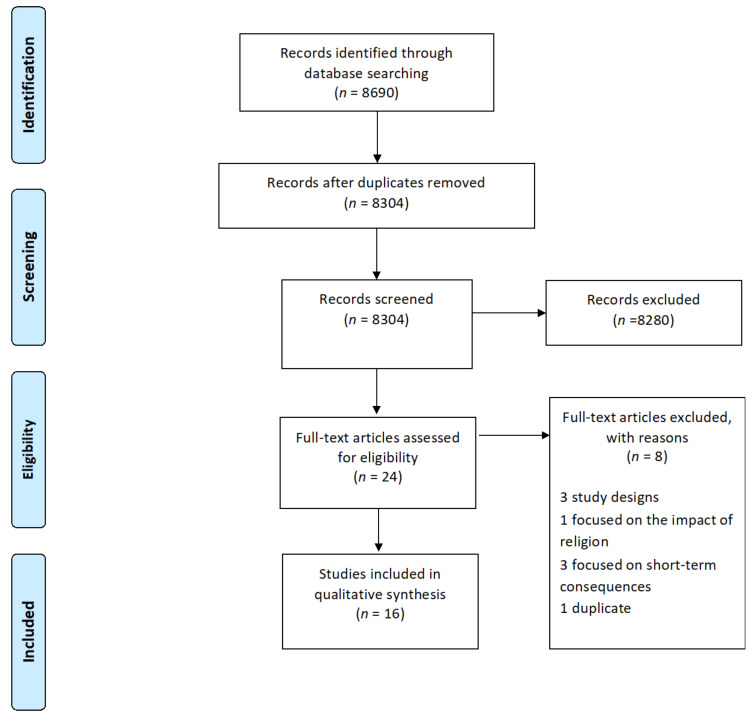
PRISMA flow diagram depicting the process of study selection for systematic review.

**Table 1 nutrients-13-04511-t001:** The main characteristics of the included studies, along with the quality assessment ordered by geographical region (South East Asia, Eastern Mediterranean, Europe, Africa, and cross-regional).

Country	Author, Year of Publication	Year of Data Collection	Age	Sample	Exposure Estimation (in Utero Ramadan Exposure) Based on	Outcome Measures Including Results	Quality Assessment Score (Max. 13)
Indonesia	Pradella and van Ewijk, 2018 [21]	1997–2008	≥15	28,489	date of birth	wheezing ↔, breathing difficulty ↓	11
Indonesia	van Ewijk et al., 2013 [22]	2000	≥18	43,649	date of birth	height ↓, weight ↓, BMI ↓	11
Indonesia	Van Ewijk, 2011 [23]	2000	≥18	29,695	date of birth	measurements of physical condition (nine-point general health scale) ↓, blood pressure ↔, pulse ↓, hemoglobin level ↔, risk of slow-healing wounds ↓, chest pain ↓	8
Indonesia	Majid et al., 2019 [24]	1993 and 2007	8–15, 18–65	Raven’s CPM: 3514, score: 3521, hours worked: 7780, earnings: 6438	date of birth	Raven’s Colored Progressive Matrices (CPM) ↓ and mathematics score ↓ (age 8–15); mean hours worked ↓ and earnings ↔ (age 18–65)	12
Indonesia	Kunto and Mandemakers, 2019 [26]	1993/1994, 1997/1998, 2000, 2007/2008, and 2014/2015.	0–19	45,246	date of birth	height-for-age Z-scores ^T^ ↓, weight-for-age Z-scores ↔, body-mass-index-for-age Z-scores ^T^ ↔	11
Indonesia	Majid, 2015 [25]	1993, 1997, 2000, 2007	6–14, 21–29	19,038	date of birth	Raven’s CPM ↓ and mathematics score ↓ (age 6–14); mean hours worked ↓ and employment ↓ (age 21–29); child labor ↓	9
Pakistan	Chaudhry and Mir, 2021 [27]	2007–2008, 2010–2011, 2013–2014, 2017–2018	0–5	179,943	date of birth	height-for-age Z-scores ^T^ ↓	12
Iran	Azizi et al., 2004 [28]	2001	4–13	191	mothers who fasted during Ramadan (questionnaire)	IQ score ↔	8
Iran	Karimi et al., 2021 [29]	2011	0–18	96,114	date of birth	height-for-age Z-scores ^T^ ↔	10
England	Almond et al., 2011 [31]	2002	7	NA	date of birth	test score on mathematics^T^ ↓, reading ^T^ ↓, writing ^T^ ↓	9
Denmark	Greve et al., 2017 [32]	1985–1995	16	NA	mothers and children immigrated from a Muslim country (≥90% Muslims)	test score on Danish ^T^ ↔, English ^T^ ↔, mathematics ^T^ ↔, science ^T^ ↔	9
Denmark	Schultz-Nielsen et al., 2016 [33]	2008	24–55	38,637	date of birth	employment ^T^ ↓, annual salary ^T^ ↔, hourly wage rate ^T^ ↔, hours of work ^T^ ↔	9
Burkina Faso	Schoeps et al., 2018 [12]	1993–2012	0–5	41,025	date of birth	under-five mortality rate^T^ ↓	12
Ethiopia	Lee et al., 2020 [30]	2000, 2005, 2011	0–4, 7–11, 15–20	21,425	month of birth	mortality rate under three months ^T^ ↓, mortality rate under one year ^T^ ↓, underweight ^T^ ↔, anemia ^T^ ↔ (age 0–4); currently enrolled in a school ^T^ ↔ (age 7–11); graduated primary school ^T^ ↔ (age 15–20)	11
USA, Iraq, Uganda	Almond and Mazumder, 2011 [34]	1989–2006	Iraq: 20–39, Uganda: 20–80	Iraq: 250,000+, Uganda: 80,000	date of birth	disability ^T^ ↓, home ownership ^T^ ↓, employment ^T^ ↔	9
International	Karimi and Basu, 2018 [35]	Varied across countries	3–4	308,879	date of birth	height-for-age Z-scores ↓	9

NA: not available; Reported association indicated upon *p* < 0.05: ↓: disadvantageous, ↔: no association; ^T^: trimester specific.

**Table 2 nutrients-13-04511-t002:** Study results according to outcome measures.

Outcome Measures among Exposed Group	Study Results
**Health outcomes**	
Body mass index (BMI)	Age-adjusted BMI difference (Δ): −0.32, 95% CI: −0.57, −0.06 [22] BMI Δ: −0.094 SD, *p* < 0.10 ^T3^ [26]
Height	Age-adjusted height Δ: −0.80 cm, 95% CI: −1.33, −0.26 [22] Height-for-age Z-score Δ: −0.105 SD, *p* < 0.05 ^T1^ [26] Height-for-age Z-score Odds ratio: 1.225, *p* < 0.001 ^T2^ [27] Height-for-age Z-score Δ: −0.091 SD, *p* < 0.01 ^T2^ [29] Height-for-age Z-score Δ: girls: 0.019, *p* = 0.613; boys: −0.073, *p* = 0.001 [35]
Weight	Age-adjusted weight Δ: −0.85 kg, 95% CI: −1.54, −0.17 [22] Weight-for-age Z-score Δ: −0.387 SD, *p* < 0.05 [26] Age-adjusted weight Δ: −0.014, *p* > 0.10 ^T1^ [30]
Disability	General disability: DID coefficient: 0.819, *p* < 0.05 ^T0^ [34] Vision: DID coefficient: 0.349, *p* < 0.10 ^T0^ [34] Hearing: DID coefficient: 0.243, *p* < 0.05 ^T0^ [34] Learning: DID coefficient: 0.250, *p* < 0.001 ^T0^ [34]
IQ scores	Mean crude full-scale IQ scores: exposed 111 ± 10, unexposed 112 ± 10 [28]
Test scores	Mathematics Δ: girls: −0.086, 95% CI: −0.158, −0.013 [24] Mathematics Δ: boys: −0.085, 95% CI: −0.151, −0.019 [24] Mathematics Δ: −0.084, *p* < 0.01 [25] DID coefficient −0.054, *p* < 0.05 ^T1^ [31] DID coefficient −0.022, *p* > 0.10 ^T0^ [32] Reading: DID coefficient −0.049, *p* < 0.05 ^T0^ [31] Writing: DID coefficient −0.051, *p* < 0.05 ^T0^ [31] English: DID coefficient −0.021, *p* > 0.10 ^T0^ [32] Raven’s CPM tests: girls: −0.092, 95% CI: −0.150, −0.03 [24] Raven’s CPM tests: boys: −0.056, 95% CI: −0.109, −0.004 [24] Raven’s CPM tests: −7.4%, *p* < 0.01 [25] Danish: DID coefficient −0.008, *p* > 0.10 ^T0^ [32] Science: DID coefficient −0.096, *p* > 0.10 ^T0^ [32]
Wheezing	Odds ratio: 1.26, *p* = 0.087 [21] among 45+ years odds ratio: 1.41, 95% CI: 0.39, 5.13 [21]
Any breathing difficulty	Odds ratio: 1.17, *p* = 0.022 [21]
General health	Age-adjusted mean Δ: −0.061, *p* < 0.01 [23]
Blood pressure	Age-adjusted mean Δ: −0.030, *p* > 0.10 [23]
Pulse pressure	Age-adjusted mean Δ: 0.939, *p* < 0.01 [23]
Hemoglobin level	Age-adjusted mean Δ: −0.054, *p* < 0.10 [23] Age-adjusted mean Δ: −0.050, *p* > 0.10 ^T1^ [30]
Risk of slow-healing wounds	Age-adjusted mean Δ: 0.047, *p* < 0.01 [23]
Chest pain	Age-adjusted mean Δ: 0.088, *p* < 0.05 [23]
Under-five mortality rate	Hazard ratio: 1.37, *p* = 0.03 ^T0^ [12]
	Hazard ratio: 1.33, *p* = 0.01 ^T1^ [12]
	Hazard ratio: 1.25, *p* = 0.05 ^T2^ [12]
Mortality under one day	Age-adjusted mean Δ: 0.005, *p* > 0.10 ^T1^ [30]
Mortality under three months	Age-adjusted mean Δ: 0.021, *p* < 0.05 ^T1^ [30]
Mortality under one year	Age-adjusted mean Δ: 0.027, *p* < 0.05 ^T1^ [30]
**Economic outcomes**	
Earnings	Δ: −0.042, 95% CI: −0.180, 0.097 [24] DID coefficient: −0.017, *p* > 0.10 ^T0^ [33]
Annual salary	DID coefficient: −0.012, *p* > 0.10 ^T0^ [33]
Home ownership	DID coefficient: −0.026 *p* = 0.027 ^T0^ [34]
Employment	Regression coefficient: −0.026, *p* < 0.05 ^T3^ [33] Regression coefficient: 0.000, *p* > 0.10 ^T0^ [34]
Hours worked	Δ: −0.075, 95% CI: −0.145, −0.016 [24] Δ: −4.7%, *p* < 0.05 [25] DID coefficient: 0.007, *p* > 0.10 ^T0^ [33]
Self-employed	Δ: 0.032, *p* < 0.05 [25]
Child labor	Δ: 0.039, *p* < 0.05 [25]
Currently enrolled in a school	Age-adjusted mean Δ: 0.024, *p* > 0.10 ^T1^ [30]
Graduated primary school	Age-adjusted mean Δ: 0.048, *p* > 0.10 ^T1^ [30]

^T0^: Exposed to Ramadan during conception; ^T1^: Exposed to Ramadan in trimester 1; ^T2^: Exposed to Ramadan in trimester 2; ^T3^: Exposed to Ramadan in trimester 3.

## Data Availability

All literature reviewed in the study was publicly available.

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
