# Peer review of "Long-Term Outcomes of in Utero Ramadan Exposure: A Systematic Literature Review"

_nutrients, 2021, doi:10.3390/nu13124511_

Round 1

Reviewer 1 Report

Overall impression:

This systematic review investigates potential associations between being in utero during the month of Ramadan and various measures of long-term health and economic well-being. Given the large proportion of muslims being exposed in one way or another to Ramadan during their early development, this investigation seems highly warranted. The paper is well-written and methodologically sound. It underpins the need for higher awareness of potential harmful effects of being exposed to fasting and/or other aspects of changed behaviors during the month of Ramadan. I have some minor comments, mostly regarding strength and weaknesses that I believe could be more transparently presented.

Abstract:

p. 1, line 35. It is not clear what is meant by age here. Age of the fetus upon exposure or age at which the outcome is measured. Please specify.

Discussion:

While the intention-to-treat principle for the comparisons is a strength in this paper, more reflection around the comparison groups seems warranted. Many of the included studies apply outcome comparisons between muslims and non-muslims. These comparisons are probably defended in the original research papers, but could have been discussed more thoroughly as strengths and limitations. Some attention should be devoted to which potential confounders the original papers have (or have not) adjusted for.

p. 4, line 145. While date of birth is an objective way of ascertaining in utero Ramadan exposure, assuming full-term birth introduces risk of bias. The authors could reflect on how the full-term assumption may have influenced findings given that prematurity is associated with the same outcomes as those reported for in utero Ramadan exposure.

p. 9, lines272-279. Could this paragraph be rephrased to make it clearer that the advice is from the compendium and not from the authors?

p. 10, lines 302-307. Please check the phrasing in these sentences.

p. 10, lines 314-315. Is it justified from this study to assert that the disadvantageous consequences are likely to be relatively small? In line with the developmental origins of health and disease theory the consequences of adverse exposures in critical phases may last for a lifetime. Besides, the muslim community is large and since between 2/3 and ¾ of all muslims may be exposed to Ramadan fasting, even small effect sizes would be important in a public health perspective.

p. 12, reference 38 is incomplete

Reviewer 2 Report

Dear Authors, I really appreciated your work. I didn't know about the potential effect of Ramadan on long terms outcomes. So good job

I only suggest to try to make the paper easier to read, simplify as much as possible concepts. 

I would suggest to introduce a sentence regarding the crucial role that maternal awareness about fetal risks has on her behavior during pregnancy, what counts most for the pregnant women is fetal wellbeing therefore as a conclusion it may be important to let the reader be aware of the importance that counseling regarding fetal risk has for maternal compliance to recommendation

therefore I suggest to read this paper and add this reference:

Quaresima P, Visconti F, Interlandi F, Puccio L, Caroleo P, Amendola G, Morelli M, Venturella R, Di Carlo C. Awareness of gestational diabetes mellitus foetal-maternal risks: an Italian cohort study on pregnant women. BMC Pregnancy Childbirth. 2021 Oct 9;21(1):692. doi: 10.1186/s12884-021-04172-y. PMID: 34627198; PMCID: PMC8502344.

Good Job!
